# China's Land Cover Fraction Change during 2001–2015 Based on Remote Sensed Data Fusion between MCD12 and CCI-LC

**Hui Wang [1], Xiaojin Wen [2], Yijia Wang [3], Liping Cai [1], Da Peng [4] and Yanxu Liu [3,*]**

1   School of Geography and Tourism, Qufu Normal University, Rizhao 276800, China;
    wh1012959497@gmail.com (H.W.); cumtcailp@126.com (L.C.)
2   School of Ecology, Resources and Environment, Dezhou University, Dezhou 253023, China;
    wenxiaojin2008@163.com
3   State Key Laboratory of Earth Surface Processes and Resource Ecology, Faculty of Geographical Science,
    Beijing Normal University, Beijing 100875, China; yijiawang27@mail.bnu.edu.cn
4   National Key Laboratory of Laser Propulsion and Applications, Space Engineering University,
    Beijing 101416, China; xiepan@thupdi.com
*   Correspondence: yanxuliu@bnu.edu.cn

**Abstract:** New types of remote sensed land cover datasets provide key evidence for understanding global environmental change. However, low data consistency makes understanding the changes unclear. China has become a hot spot of land cover change in the world due to climate change and a series of human measures, such as ecological engineering, land consolidation, and urbanization. However, due to the inconsistencies in interpretation of signs and thresholds, the understanding of yearly-continued land cover changes in China is still unclear. We aim to produce China's land cover fraction dataset from 2001 to 2015 by weighted consistency analysis. We compare the Moderate-resolution Imaging Spectroradiometer land cover dataset (MCD12Q1), the Climate Change Initiative Land Cover (CCI-LC) datasets, and a new land cover fraction dataset named China-LCF$_{MCD-CCI}$, produced with a 1 km resolution. The obvious increased forest areas only accounted for 4.6% of the total forest areas, and were mainly distributed in northeast China. Approximately 75.8% of the grassland and shrubland areas decreased in size, and these areas were relatively concentrated in northeast and south China. The obvious increased areas of cropland (3.7%) were equal to the obvious decreased areas (3.6%), and the increased cropland areas were in northwest China. The change in bare land was not obvious, as the obvious increased areas only accounted for 0.75% of the bare land areas. The results not only prove that the data fusion of the weighted consistency method is feasible to form a land cover fraction dataset, but also helps to fully reveal the trends in land cover fraction change in China.

**Keywords:** land cover; accuracy analysis; data fusion; weight consistency; uncertainty

## 1. Introduction

Land cover is a dynamic variable that changes rapidly across the globe [1,2]. Therefore, high-quality and timely land cover information is important for addressing global environmental change issues [3]. Global land cover datasets provide fundamental information on land surfaces and are important variables in environmental research [2,4,5]. With the continuous development of remote sensing technology, various global and regional land cover products have emerged, such as International Geosphere-Biosphere Programme, Data and Information Systems (IGBP-DIS) land cover [6], Global Land Cover 2000 (GLC2000) [7], Globeland30 [8], and so on. However, due to the differences in sensor types, spatial and temporal resolution, interpretation marks, classification methods, and thresholds, the datasets have uncertainty and poor comparability [9,10]. To identify the relative certain trends in land cover change, there is an urgent need for interactive verification between multi-source data.

China is experiencing both fast climate change and rapid urbanization, which have brought about food security issues as well as environmental security issues [11]. Moreover, China has implemented a series of policies influencing land cover change. China's ecological engineering programs, such as the Grain for Green Program, have increased vegetation coverage and changed the forest cover across tens of millions of hectares [12,13]. China's land consolidation has focused on two core objectives: replenishing arable land and improving agricultural infrastructure [14,15]. China's land consolidation has supplied 2.8 million hectares of cropland and promoted the sustainable development of rural areas. From 2000 to 2010, the total cropland decreased by 0.7% [16]. To slow down the loss of farmland caused by urbanization, China has implemented the strategy of balancing cultivated land occupation and compensation, which requires the same proportion of cultivated land expansion to compensate for the loss in cultivated land [17]. In addition, China has implemented a number of land conservation projects, including soil and water conservation projects and grassland ecological conservation programs [18]. Therefore, as China has become a prominent area of global land cover change through a series of land use policies, it is urgent to identify the land cover change in China through interactive verification of more data sources in order to form a more rational judgement of land cover change trends and uncertain regions.

Currently, there are several remote sensed land cover datasets that can be applied in China. Although MCD12Q1 has a relatively low spatial resolution (500 m), it has a continuous time series from 2001 to 2015. In addition, MCD12Q1 contains a variety of classification schemes, including IGBP classification, the University of Maryland (UMD) classification, leaf area index (LAI) classification, Biome-biogeochemical cycle (BGC) classification, plant functional type (PFT) classification, and FAO (Food and Agriculture Organization) -land cover classification system land cover (LCCS). The Climate Change Initiative Land Cover (CCI-LC) dataset has a spatial resolution of 300 m and was classified into 22 types of land cover [19]. The greatest advantage of CCI-LC is that it had a continuous time series from 1992 to 2015. Finer than these global datasets, China's land cover datasets are often at a 30 m resolution. China land cover is a high-resolution dataset based on the land cover data of the ten-year change of the environment in China and the classified data were accurately matched with Landsat Thematic Mapper (TM) images and then classified via deep learning (http://data.casearth.cn/). The Global Land Cover Fine Surface Covering 30 (GLC-FCS30) dataset also has a high resolution of 30 m and combines a multitemporal classification model and a spatial-temporal spectral library [20]. The spatial-temporal spectral library was developed by CCI-LC and the MODIS Version 6 Nadir bidirectional reflectance distribution function adjusted reflectance (NBAR) product (MCD43A4) [20]. Globeland30, with an overall accuracy of over 80%, is the first work to map global land cover at a 30 m resolution [21]. Globeland30 holds more than 20,000 images, mainly from TM, enhanced thematic mapper plus (ETM+), and the Chinese HJ-1 satellite images, so that it can provide details on land cover patterns [22]. However, there is currently no unified view on the changing tendency of China's land cover because continuous time series datasets are often at the global scale, and there is a lack of evidence on whether one dataset is more reliable than others for different areas of China.

Data consistency analysis is a conventional method for evaluating land cover datasets that usually reflects the quality and practicality of the datasets better than the overall accuracy [23]. The commonly used methods of land cover consistency analysis include the sample point verification method, the superposition analysis method, and the Boolean analysis method [5,24–27]. Fritz et al. [25] used three independent validation works to evaluate the consistency of the existing 1 km dataset. Liang et al. [27] used two sets of sample points collected from the Arctic region to assess the accuracy of CCI-LC, Global Land Cover by the National Mapping Organization (GLCNMO), Globeland30, and MODIS. Bai et al. [24] found that MODIS LC has the highest consistency compared with the Global Land Cover Characterization (GLCC), University of Maryland land cover product (UMd), Global Land Cover 2000 project data (GLC2000), and GLOBCOVER land cover (GlobCover)

products via superposition analysis. Among the abovementioned methods, the weighted consistency analysis method can overcome the shortcomings associated with the difficulty of data acquisition in the sample verification method and the lack of a continuous time series in the superposition analysis method [24,27].

Although the Chinese products of land cover datasets may have higher precision than the global ones, most of the land cover datasets in China are category variables and lack of fraction variables, and they are lack of continuous time series. To obtain the spatial consistency of the dataset with long time series is a reasonable approach for improving the spatiotemporal accuracy of China's land cover datasets. Thus, the hypothesis of this study should be forming a land cover fraction dataset in China based on the weighted consistency data fusion method.

In order to produce a set of land cover fraction datasets in continuous year series, three research steps were designed: (1) to evaluate the consistency of multi-source land cover data by various methods; (2) to fuse MCD12 and CCI-LC data through a consistency analysis to form a time series land cover fraction map; and (3) to analyse the tendency of land cover change in China from 2001 to 2015.

## 2. Materials and Methods

### 2.1. The Land Cover in China

Around 75% of China's land is covered with forest, grassland, and cropland (Figure 1) [28]. Remote sensed evidence showed the land cover change in China was significant [29]. The urban area has rapidly expanded with the economic development in China [30]. The forest decreased during the last decade of the 20st century, and increased in the first decade of the 21st century, while the grassland, wetland, and unused land continued decreasing in 21st century [31].

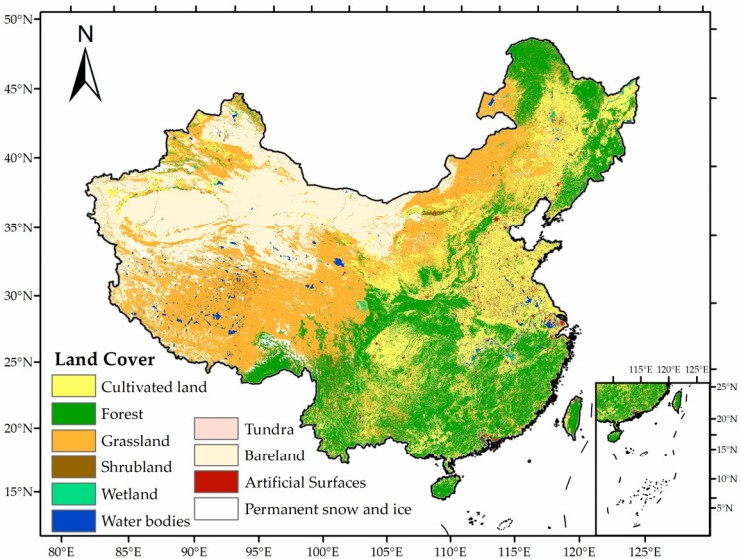

**Figure 1.** The main land cover of China (Globeland30 product of 2010).

### 2.2. Data Source and Pre-Processing

In this study, we used five land cover datasets for the consistency analysis and data fusion, and then we used different land cover type datasets to validate the results (Table 1).

(1) CCI-LC with the United Nations (UN) Land Cover Classification System (LCCS) is created by the European Space Agency [19] and is available at an annual scale.

(2) MCD12Q1 with the IGBP land cover classification scheme is produced by the United States Geological Survey [32], and it is also a dataset for successive years.

(3) China land cover is developed by the Institute of Remote Sensing and Digital Earth, Chinese Academy of Sciences, and it includes eight land cover types [33].

(4) GLC-FCS30, with a fine classification scheme that includes 22 land-cover types, is generated by the Institute of Remote Sensing and Digital Earth, Chinese Academy of Sciences [20].

(5) Globeland30 was developed by the National Geomatics Center of China and includes 10 land cover types [8].

(6) The Global Artificial Impervious Area (GAIA) is mapped by the Department of Earth System Science, Tsinghua University using the full archive of 30 m resolution Landsat images on the Google Earth Engine platform [34].

(7) The global cropland-percentage-map is produced by the Institute of Agricultural Resources and Regional Planning, Chinese Academy of Agricultural Sciences, and it can be used for producing the global dataset of agricultural production for 2010 [35].

(8) Globeland30-WTR2010 is produced by the National Geomatics Center of China and uses three types of water body extraction algorithms to automatically extract land surface water [36].

(9) The Hansen-GFC (global forest change) is mapped by the University of Maryland, and it can observe global forest loss and gain from 2001 to 2012 [37].

As the datasets have different resolutions from 30 m to 500 m, the spatial resolution is unified to 1 km using the nearest neighbour method in the categorical variable consistency analysis (Figure 2). Since the datasets have different classification systems, e.g., CCI-LC uses the LCCS systems with 22 land cover types and MCD12Q1 uses IGBP systems with 17 types, we re-classed the land cover types into 8 or 9 types (Table 2) (China land cover does not include shrubland).

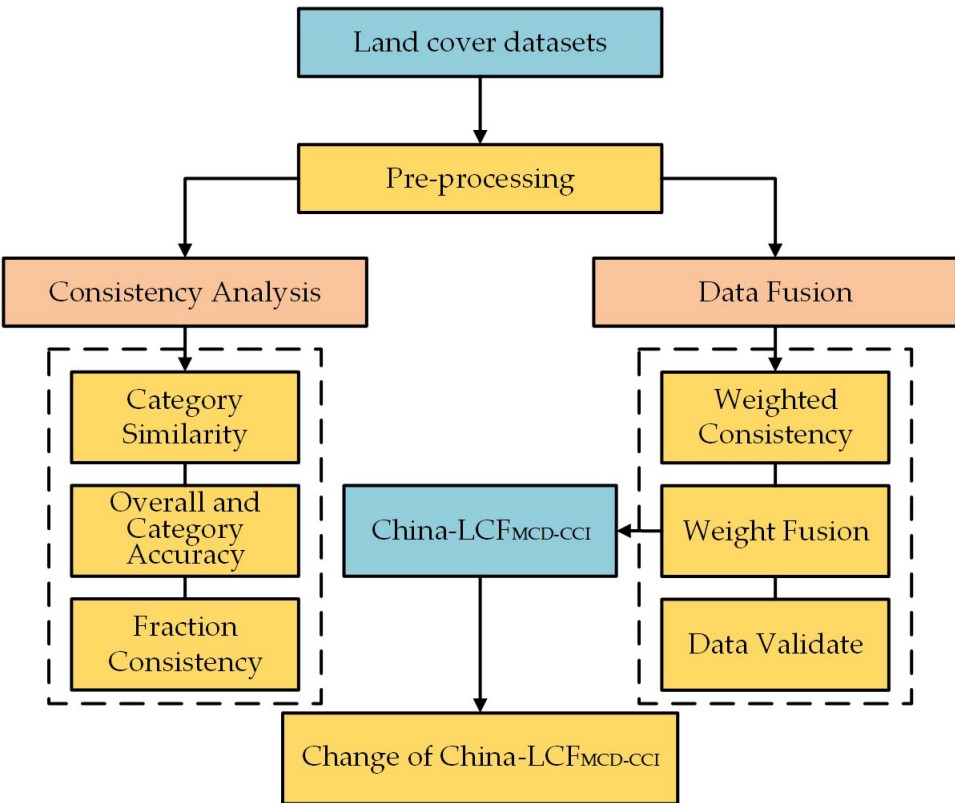

**Figure 2.** Flowchart of the produce of land cover fraction dataset in China (Drawing by Origin 2019).

<div align="center">

**Table 1.** Characteristics of multi-source land cover datasets.

</div>

| Dataset | Resolution (m) | Time | Classification Method | Source |
|---|---|---|---|---|
| CCI-LC | 300 | 2001–2015 | Unsupervised classification | https://www.esa-landcover-cci.org/ |
| China land cover | 30 | 2005–2010 | Deep learning | http://data.casearth.cn/ |
| GAIA | 30 | 2015 | Exclusion-inclusion framework | http://data.ess.tsinghua.edu.cn/ |
| GLC-FCS30 | 30 | 2015 | Spatial-temporal spectral library | http://data.casearth.cn/ |
| Global-cropland-percentage-map | 500 | 2010 | Self-adapting Statistics Allocation Model | https://doi.org/10.7910/DVN/ZWSFAA |
| Globeland30 | 30 | 2000–2010 | Pixel-Object-Knowledge | http://www.globeland30.com |
| Globeland30-WTR2010 | 30 | 2010 | Decision tree | http://www.geodoi.ac.cn |
| Hansen-GFC | 30 | 2000 | Decision tree | http://earthenginepartners.appspot.com/science-2013-global-forest |
| MCD12Q1 | 500 | 2001–2015 | Decision tree | https://ladsweb.modaps.eosdis.nasa.gov/ |

<div align="center">

**Table 2.** Merged classification system.

</div>

| Type | CCI-LC | MCD12Q1 (IGBP) | Globeland30 | GLC-FCS30 | China Land Cover |
|---|---|---|---|---|---|
| Forest | 40/50/60/61/62/70/71/72/80/81/82/90/100/160/170 | 1/2/3/4/5 | 20 | 12/50/60/61/62/70/71/72/80/81/82/90 | 1 |
| Grassland | 110/130 | 8/9/10 | 30 | 11/130 | 2 |
| Shrubland | 120/121/122 | 6/7 | 40 | 120/121/122 | / |
| Cropland | 10/11/12/20/30 | 12/14 | 10 | 10/20 | 6 |
| Wetland | 180 | 11 | 50 | 180 | 3 |
| Water | 210 | 17 | 60 | 210 | 4/5 |
| Construction | 190 | 13 | 80 | 190 | 7 |
| Bare land | 140/150/151/152/153/200/201/202 | 16 | 90 | 140/150/152/153/200/201/202 | 9 |
| Permanent snow and ice | 220 | 15 | 100 | 220 | 10 |

*2.3. Consistency Analysis*

2.3.1. Relative Consistency Analysis

The relative consistency analysis includes the category similarity analysis, the overall and category accuracy analysis, and fraction consistency analysis, which are the spatial statistical results of any two datasets in four land cover datasets [38]. Since 2010 was the only year of overlap among all land cover datasets, we only analysed the relative consistency of the land cover datasets in 2010.

Category similarity is obtained by summing the areas of different land types and calculating the correlation coefficients of the different land type areas of the two land cover datasets. The formula is:

$$R_i = \frac{\sum_{i=1}^{n}(X_i - \overline{X})(Y_i - \overline{Y})}{\sqrt{\sum_{i=1}^{n}(X_i - \overline{X})^2 \sum_{i=1}^{n}(Y_i - \overline{Y})^2}} \tag{1}$$

where $R_i$ is the correlation coefficient (category similarity) of two land cover datasets, $i$ is the land cover type, $X_i$ represents the area of type $i$ in land cover dataset $X$, $Y_i$ refers to the area of type $i$ in land cover dataset $Y$, $\overline{X}$ is the average of the total area of all types in

dataset $X$, $\overline{Y}$ refers to the average of the total area of all types in dataset $Y$, and $n$ is the total number of land cover types. In this part, $n$ is 8 because the China land cover dataset does not include shrubland.

Overall and category accuracy analysis indicate the overall accuracy and user accuracy among each land cover dataset. Recently, confusion matrices have become a common method for evaluating the accuracy of land cover datasets [39,40]. The confusion matrix can describe the accuracy index of land cover datasets and obtain the corresponding information for each pixel in each dataset [9]. A confusion matrix includes the overall accuracy (OA), user accuracy (UA), producer accuracy (PA), and Kappa coefficient. OA and Kappa represent the overall accuracy, and PA represents the category accuracy. The formulas of each index are as follows:

$$\text{OA} = \frac{\sum_{i=1}^{r} x_{ii}}{N} \times 100\%, \tag{2}$$

$$\text{PA} = \frac{x_{ii}}{x_{+i}} \times 100\%, \tag{3}$$

$$\text{Kappa} = \frac{N \cdot \sum_{i=1}^{r} x_{ii} - \sum_{i=1}^{r} (x_{i+} \cdot x_{+i})}{N^2 - \sum_{i=1}^{r} (x_{i+} \cdot x_{+i})}, \tag{4}$$

where $N$ is the total number of pixels, $r$ is the number of types ($r$ = 8), $x_{ii}$ represents the pixels of the correct classification, $x_{+i}$ refers to the number of pixels of type $i$ in the reference dataset, and $x_{i+}$ stands for the number of pixels of type $i$ in the evaluated dataset.

Fraction consistency (FC) analysis can calculate the fractions of different land types in a larger-scale grid. We used the "Block Statistics" tool of ArcGIS 10.2 to analyse the percentage of each land cover type in 1 km grids. For example, if we set the forest of Globeland30 to 1, and set the rest of the land types to 0, and then perform a block statistical analysis, we can obtain the proportion of the Globeland30 forest on the 1 km grid. After the block statistics of all land types in different land cover datasets, a fraction consistency analysis can be performed. The formula is as follows:

$$\text{FC}_{ij} = \frac{\sum_{j}^{4} pro_{ij}}{4}, \tag{5}$$

where $\text{FC}_{ij}$ represents the fraction consistency of land cover type $i$ in dataset $j$ and $pro_{ij}$ refers to the fraction of land cover type $i$ in dataset $j$. The number 4 represents four land cover datasets: CCI-LC, MCD12Q1, China land cover, and Globeland30.

### 2.3.2. Weighted Consistency Analysis

In the weighted consistency analysis, we chose CCI-LC and MCD12Q1 as the two datasets for weighting. Because the time distribution of the datasets is not consistent, we only selected datasets of the same year for the analysis. For example, for the CCI-LC dataset, we chose Globeland30 (2000), China land cover (2005), Globeland30 (2010), China land cover (2010), and GLC-FCS30 (2015) to analyse the weighted consistency. Then, the weight was calculated, which represents the proportions of the CCI-LC and MCD12Q1 datasets in the fractions of the different land cover types in the 1 km grid. The weight was calculated on the basis of data fusion, and its calculation formula is as follows:

$$W_{i\text{-}CCI} = \frac{\sum_{k} \text{FC}_{i\text{-}CCI\text{-}k}}{4}, \tag{6}$$

$$W_{i\text{-}MCD} = \frac{\sum_{k} \text{FC}_{i\text{-}MCD\text{-}k}}{3}, \tag{7}$$

where $W_{i\text{-}CCI}$ is the weight of the CCI-LC dataset in land use type $i$ and $W_{i\text{-}MCD}$ is the weight of the MCD12Q1 dataset in land use type $i$. In Formula (6), $k$ equals 2000, 2005, 2010, or 2015, and in Formula (7), $k$ equals 2005, 2010, or 2015. $\text{FC}_{i\text{-}CCI\text{-}k}$ and $\text{FC}_{i\text{-}MCD\text{-}k}$ refer

to the fraction consistency of land cover type *i* in the CCI-LC or MCD12Q1 dataset in year *k*, respectively.

### 2.4. Data Fusion

We fused CCI-LC and MCD12Q1 according to their weights. Due to the small shrubland area in China and the lack of a shrubland cover type in the China land cover dataset, we merged grassland and shrubland for data fusion. We also overlayed all of the land cover types to better show the probability distribution of various land cover types. The formula for data fusion is as follows:

$$\text{China-LCF}_{\text{MCD-CCI}} = \frac{\text{FC}_{i\text{-}CCI} \cdot W_{i\text{-}CCI} + \text{FC}_{i\text{-}MCD} \cdot W_{i\text{-}MCD}}{\sum_{i=1}^{n} W_{i\text{-}CCI} \cdot \text{FC}_{i\text{-}CCI} + \sum_{i=1}^{n} W_{i\text{-}MCD} \cdot \text{FC}_{i\text{-}MCD}}, \tag{8}$$

where $\text{FC}_{i\text{-}CCI}$ and $\text{FC}_{i\text{-}MCD}$ are the FC of land cover type *i* in CCI-LC and MCD12Q1, respectively, *n* is the total number of land cover types, and $W_{i\text{-}CCI}$ and $W_{i\text{-}MCD}$ are the weights of land cover type *i* in CCI-LC and MCD12Q1, respectively.

### 2.5. Contrast with High Resolution Dataset

We selected four different land cover type datasets to validate China-LCF$_{\text{MCD-CCI}}$, including GAIA, the Global-cropland-percentage-map, Hansen-GFC and Globeland30-WTR2010. Because these datasets are the most reliable high-resolution datasets with full coverage available in China. Similarly, the block statistical analysis conducted on four datasets. We also compared the four datasets with CCI-LC and MCD12Q1. The values of different datasets were obtained by randomly taking 10,000 points for each land cover type, and then the coefficient of determination ($R^2$) and root mean square error (RMSE) between the datasets were calculated. If the $R^2$ values between the China-LCF$_{\text{MCD-CCI}}$ and the other four datasets was higher than the $R^2$ values between MCD12Q1, CCI-LC and the other four datasets, it meant that our results were closer to the results on the fine resolution scale and had a higher credibility than those of CCI-LC and MCD12Q1.

In addition, we calculate the relative deviation (RD) to represent uncertainty of the results, and the formula is as follows:

$$RD_i = \frac{\text{China-LCF}_{\text{MCD-CCI-}i} - \overline{S}}{\overline{S}} \tag{9}$$

where $RD_i$ are the relative deviation of land cover type *i*. $\overline{S}$ are the average between China-LCF$_{\text{MCD-CCI}}$ and GAIA, the Global-cropland-percentage-map, Hansen-GFC and Globeland30-WTR2010, respectively.

### 2.6. Change of Land Cover Fraction

We analysed the temporal and spatial variations in the land cover fraction in China and the different sample regions. On the time scale, we performed a linear regression analysis on China-LCF$_{\text{MCD-CCI}}$ from 2001 to 2015 and calculated the slope of the regression equation. From the slope map, we can find changes in the fraction of land cover in China from 2001 to 2015 with obvious change areas with |Slope| > 0.02. On the spatial scale, to show the changes of China-LCF$_{\text{MCD-CCI}}$ for the different land cover types from 2001 to 2015, we selected four sample regions from four different land cover types (Forest, Grassland and Shrubland, Cropland, Bare land) to analyse the changes of the average values. We performed a Mann–Kendall test on the data to calculate its significance (tested by MATLAB 2019a). If |Z-score| is greater than or equal to 1.28, it passes the significance test at 10% level; if |Z-score| is greater than or equal to 1.64, it passes significance test at 5% level; if |Z-score| is greater than or equal to 2.32, it passes significance test at 1% level.

## 3. Results

### 3.1. Consistency of the Datasets

　　The correlation coefficients of the five land cover datasets were quite different, ranging from 0.57 to 0.98 (Figure 3a). The category similarity (correlation coefficient) between Globeland30 and CCI-LC was the highest, while the category similarity between China land cover and MCD12Q1 was the lowest (Figure 3a). The correlation coefficients between CCI-LC and MCD12Q1 was relatively high (0.84). The mean value of the category similarity between CCI-LC and the other three datasets was 0.86, and the mean value between MCD12Q1 and the other three datasets was 0.78. Therefore, these results show that Globeland30 has the best category similarity among different land cover types and that China land cover has the worst category similarity.

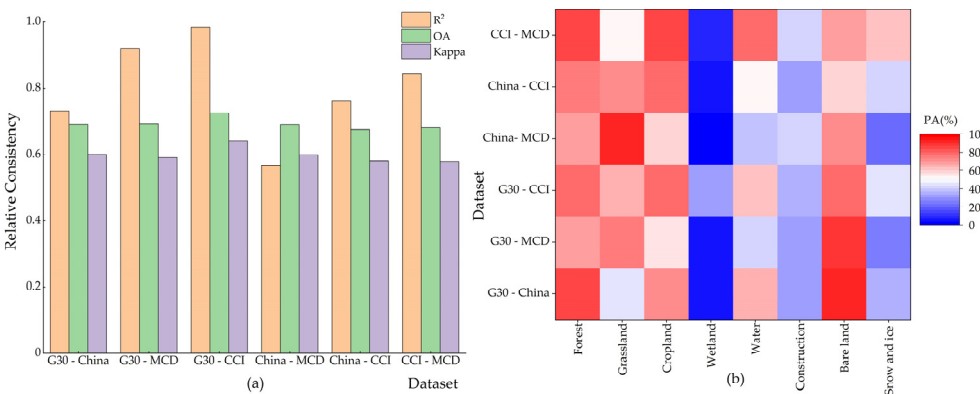

**Figure 3.** (**a**) R$^2$ (coefficient of determination), overall accuracy (OA), Kappa and (**b**) producer accuracy (PA) of different land cover products in 2010. (G30: Globeland30; China: China land cover; MCD: MCD12Q1; CCI: Climate Change Initiative Land Cover (CCI-LC)).

　　The OA between Globeland30 and CCI-LC was the highest (0.73) (Figure 3a). The second highest OA was between Globeland30 and MCD12Q1 (0.69). The OA between Globeland30 and China land cover ranked third (0.69). Meanwhile, the lowest OA was between China land cover and CCI-LC (0.68). The OA between Globeland30 and the other three datasets was the highest, and the OA between China land cover and the other three datasets was the lowest. There was little difference in the OA between the different datasets, and it was always more than 0.60. Another indicator for assessing the overall consistency is Kappa, which has similar characteristics to OA. The difference in Kappa between the four datasets was not large. The highest Kappa was also found between Globeland30 and CCI-LC (0.64), and the lowest Kappa was between CCI-LC and MCD12Q1 (0.58). The highest Kappa values was between Globeland30 and the other three datasets, and the lowest Kappa vales was between China land cover and the other three datasets. Based on the OA and Kappa, Globeland30 has the highest overall accuracy, and MCD12Q1 has the lowest overall accuracy.

　　We also obtained the mean values of PA in different land cover types across the four datasets (Figure 3b). The mean value of PA between CCI-LC and MCD12Q1 was the highest (60.4%), and the lowest mean value of PA was between Globeland30 and MCD12Q1 (48.0%). The PA between the four datasets varied greatly with different land cover types. We found that the mean value of PA in forests was the highest (76.6%) and the mean value of PA in wetlands was the lowest (9.38%). The spectral characteristics of wetlands are not obvious, and wetlands are easy to mix with other land objects in terms of their spatial distribution [9]. The PA or category accuracy of wetlands was lower than that of other land cover types. The PA in cropland, water, construction, and permanent snow and ice was the highest between CCI-LC and MCD12Q1 compared with the other datasets. Similarly, the PA in forest and bare land was the highest between Globeland30 and China land cover. In addition, the highest PA in grassland was found between China land cover and MCD12Q1,

and the highest PA in wetland was found between Globeland30 and CCI-LC. Overall, the category accuracy of CCI-LC was the greatest in most of the land cover types.

The FC of different land cover types between the four datasets in 2010 had different distributions (Figure 4). China's forests are mainly distributed in the northeast and southern regions. The regions with a higher FC were in the eastern mountains in the northeast, the Hanzhong Basin and the southern wing of the Eastern Himalayas. In addition, the eastern part of Taiwan has a high FC. Grassland is widely distributed in most areas of China, where FC was high in the Qinghai-Tibet Plateau, in northern Xinjiang and in the Inner Mongolia Plateau and the FC was relatively low in southern China. The cropland of China is mainly distributed in eastern coastal regions. The Northeast Plain, Sanjiang Plain, North China Plain and Sichuan Basin had a higher FC for cropland than the other regions. Bare land is mainly distributed in western China, including Xinjiang, Tibet, Qinghai and Inner Mongolia. Among these areas, most areas of Xinjiang and western Inner Mongolia had a higher FC than Tibet and Qinghai. Because bare land had the highest average value and grassland had the lowest, bare land had the best fraction consistency and grassland had the worst.

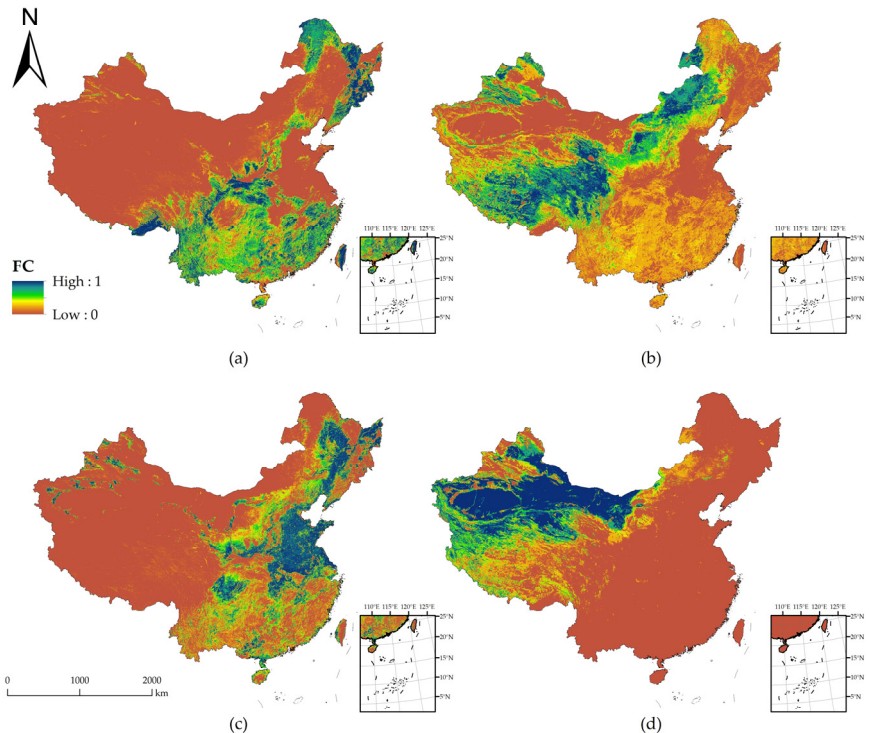

**Figure 4.** FC of different land cover datasets in (**a**) forest; (**b**) grassland; (**c**) cropland; and (**d**) bare land in 2010. (FC: fraction consistency).

### 3.2. Data Fusion Product

We got the results of data fusion (Figure 5) and counted the average value of China-LCF$_{MCD-CCI}$ in China among different land cover types from 2001 to 2015 (Figure 6). The average values of grassland and shrubland were significantly larger than the other land cover types (Figure 6). In addition, the interannual variation in the average value in China was stable. Since China covers a large area, we selected four sample regions for different land cover types to describe the characteristics of the China-LCF$_{MCD-CCI}$ and got the changes in the average values for the sample regions from 2001 to 2015 (Figure 7).

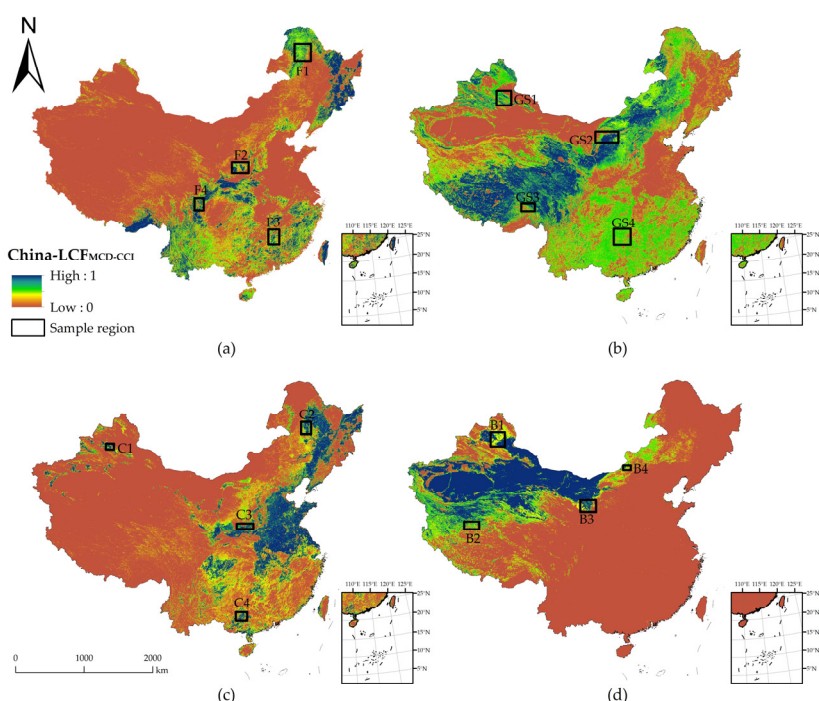

**Figure 5.** (**a**) F: forest; (**b**) GS: grassland and shrubland; (**c**) C: cropland; and (**d**) B: bare land sample regions and China-LCF$_{MCD-CCI}$ (land cover fraction) results in 2010. The numbers represent the number of the sample region.

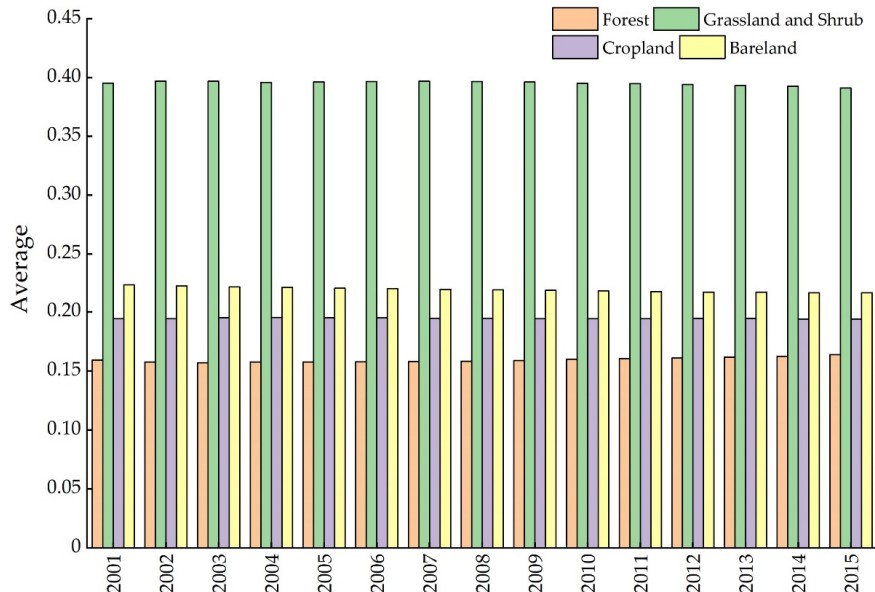

**Figure 6.** The average value of China-LCF$_{MCD-CCI}$ in different land cover types.

In forests, F1 stood for the Greater Khingan Mountains, which is an important forest region in China, and its average value was increasing and fluctuating. F2 represented the Loess Plateau, and its variation was the most stable of the four forest sample regions. F3 denoted the Jiangnan Hills, and the average value in this region was decreasing. F4 represented the Sichuan Basin, and the average values in this region were higher than those in the other sample regions, and showed a decreasing trend. The change rate of F1 was the highest, and F2 was the lowest. In grassland and shrubland, the average of all four sample regions showed slightly increasing trends. GS1 represented the Junggar Basin in northern Xinjiang, GS2 represented the Inner Mongolia Plateau, GS3 represented the southeastern

part of the Qinghai-Tibet Plateau and GS4 represented the Jiangnan Hills. In the four land cover types, the change in the average values in cropland was the largest. C1 denoted the Junggar Basin and had a growth rate of 38.8%. C2 represented the Northeast China Plain, and it had a distinct increasing tendency. C3 stood for the Loess Plateau, and its variation was the most stable among the four sample regions of cropland. C4 represented the Yunnan-Kweichow Plateau, and it was the only one of the four sample regions to show a decreasing trend. Since the distribution of bare land is mainly concentrated in northwest China, our sample regions were also distributed in this area. B4 denoted the Inner Mongolian Plateau, and its average value had an obvious decrease in 2001–2013. B1 represented the Junggar Basin, B2 represented the southern Kunlun Mountains, and B3 represented the Hexi Corridor. The variation trends of B1, B2, and B3 were relatively stable. The changes in the grassland and shrubland were the most stable, while the changes in cropland were the most volatile (Figure 7). The result of Mann–Kendall showed that most of our result passes significance test at 1% level.

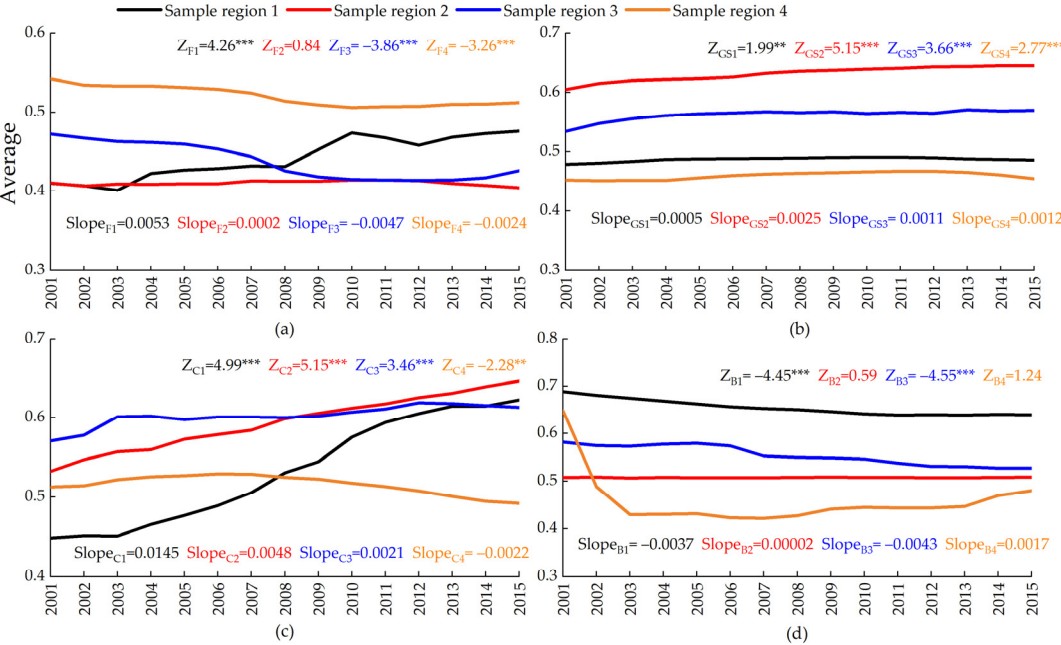

**Figure 7.** Compare the average value of China-LCF$_{MCD-CCI}$ in four sample regions. (**a**) Forest; (**b**) grassland and shrub; (**c**) cropland; (**d**) bare land (Z: Z-score of Mann–Kendall test; slope: slope of Mann–Kendall test; *: Significance level (**: 5%, ***: 1%)).

### 3.3. The Change of China-LCF$_{MCD-CCI}$

Overall, the $R^2$ values between China-LCF$_{MCD-CCI}$ and Hansen, Crop, GAIA, and WTR-2010 were greater than those between MCD12Q1 and CCI-LC, and the other datasets, while the RMSEs were less than those between the MCD12Q1 or CCI-LC and the other datasets (Figure 8). This indicated that China-LCF$_{MCD-CCI}$ was reliable and could have effectively reflected land-cover fractions in China and its changes from 2001 to 2015. The $R^2$ between GAIA and MCD12Q1 was the lowest (0.38), but the $R^2$ between GAIA and China-LCF$_{MCD-CCI}$ was relatively high (0.51). Although the $R^2$ values between the datasets were lower than 0.6, China-LCF$_{MCD-CCI}$ was a fusion of MCD12Q1 and CCI-LC and improved the accuracy of MCD12Q1 or CCI-LC. Due to the relatively low resolution of MCD12Q1 and CCI-LC, the spatial heterogeneity of their fraction on the 1 km grid cannot be clearly reflected, which is improved by China-LCF$_{MCD-CCI}$. The national average for forest uncertainty was highest (16.5%) and the national average for cropland uncertainty was lowest (1.46%). The national average for water uncertainty was 4.27% and the national

average for construction uncertainty was 7.23%. The uncertainty showed that China-LCF$_{\text{MCD-CCI}}$ were reliable.

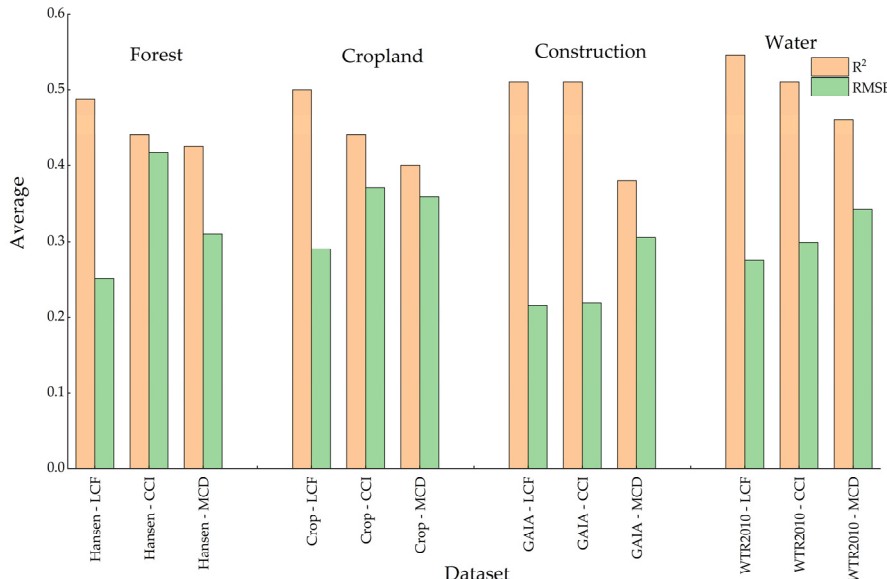

**Figure 8.** Comparison of R$^2$ and root mean square error (RMSE) among China-LCF$_{\text{MCD-CCI}}$, MCD12Q1, CCI-LC, and different land cover type datasets. (Hansen: Hansen-GFC (global forest change); LCF: China-LCF$_{\text{MCD-CCI}}$; CCI: CCI-LC; MCD: MCD12Q1; crop: global cropland percentage map; GAIA: Global Artificial Impervious Area; WTR2000: Globeland30-WTR2010).

We analysed the slope of China-LCF$_{\text{MCD-CCI}}$ change from 2001 to 2015 (Figure 9). For forests, the area with a large change (0.02 < |Slope| < 0.1) was small, and the distribution was not concentrated. The areas with obvious increases (slope between 0.02 and 0.1) only accounted for 4.6% of the total forest area, and the areas with obvious decreases (slope between −0.1 and −0.02) only accounted for 2.2%. Areas with an obvious decrease in terms of forest were distributed in the Hanzhong Basin and its southern region. The greater Khingan Mountains was an area in which the forest area obviously increased. Similar to the changes in forest area, the areas with obvious changes in grassland and shrubland were scattered and not concentrated. The area in which grassland and shrubland obviously increased (3.6%) was smaller than the area in which grassland and shrubland obviously decreased (4.6%). Most areas in China have experienced a decrease in grassland and shrubland, but the areas with relatively concentrated decreases were the Greater Khingan Mountains and southern China (Figure 9b). The area of grassland and shrubland obviously increased in Sichuan Basin. For cropland, the areas with obvious increases (3.7%) and decreases (3.6%) were (basically) essentially equal. Among them, the areas with obvious decreases were Shandong Peninsula and Sichuan Basin, and the areas with obvious increases were Junggar Basin and the Loess Plateau. For bare land, the areas with little change (slope between −0.02 and 0.02, except 0) accounted for 85.8% of the bare land. The areas with obvious decreases in bare land area were distributed in Junggar Basin and Hetao. The areas with obvious increases in bare land accounted for only 0.75% of the total bare land. In China as a whole, although the change rates of the average values from 2001 to 2015 in different land cover types were low, forests have increased, while grassland and shrubland, cropland, and bare land have decreased.

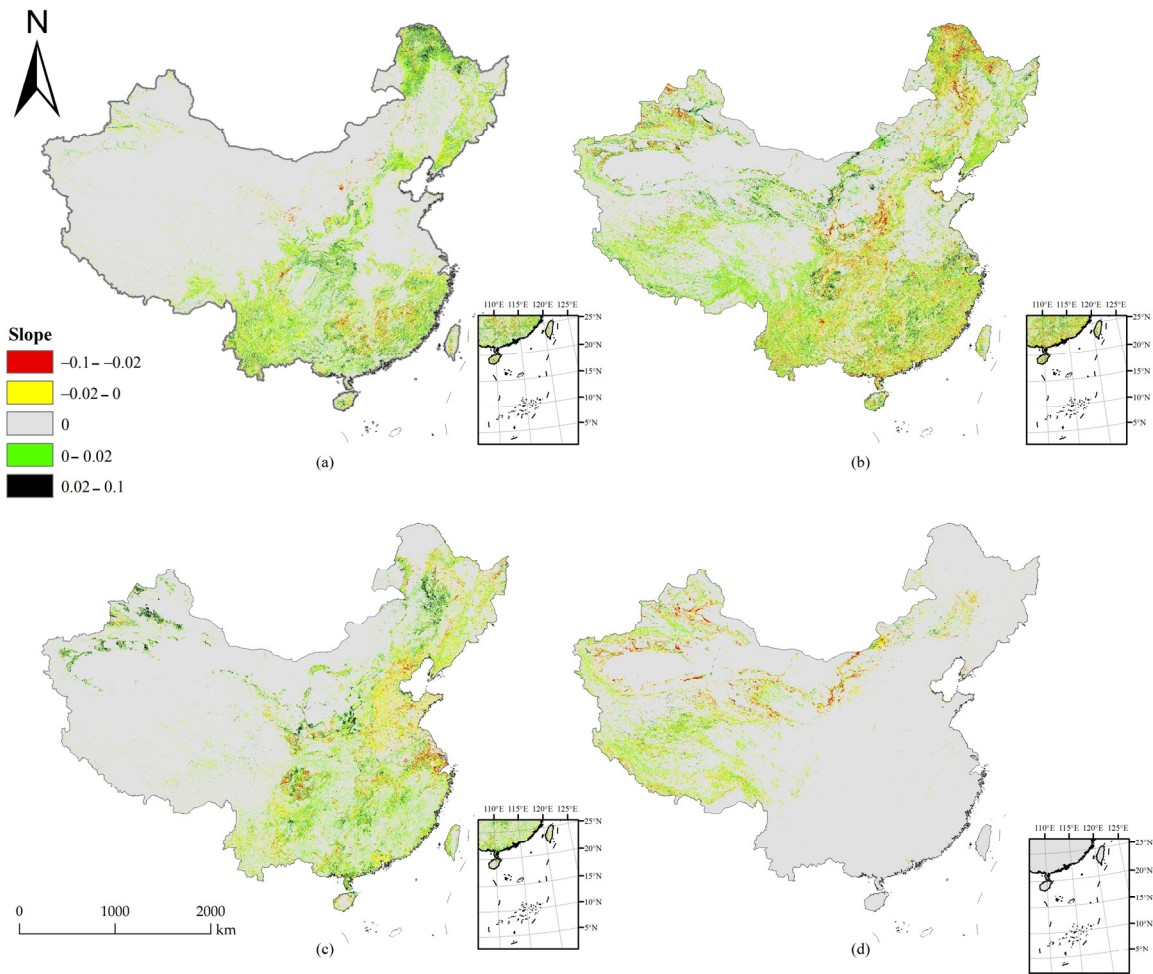

**Figure 9.** The slope of the China-LCF$_{MCD-CCI}$ change in 2001–2015. (**a**) Forest; (**b**) grassland and shrub; (**c**) cropland and (**d**) bare land.

## 4. Discussion

We compared the change rate and change map in the case study area with the previous findings (Table 3). Huang et al. [41] used all Landsat images from the Google Earth Engine to create a dynamic map of the major land cover from 1985 to 2015 in Beijing, and they found that the percentage of forest gain (21.8%) was greater than the percentage of forest lost (2.1%) because the land was revegetated. In our study, the forest in Beijing increased from 2001 to 2015, and the rate of increase was 10.06%. The annual change rate was similar to that of Huang. Zhu et al. [42] incorporated the impact of the land cover change into the assessment of greenness trends in Guangzhou from 2000 to 2014, and found that the area forest loss (3.4 ± 0.5%) was slightly larger than the area of forest gain (2.4 ± 0.5%). In our results, the forest in Guangzhou increased from 2001 to 2014, and the change rate (28.6%) was obviously higher than 1.0 ± 0.5%. The areas where forests increase are mainly in the northern and central parts of Guangzhou (Table 3c). Cao et al. [43] analysed the urban expansion in Xishuangbanna and its impact on the land use pattern. They concluded that the forest area in Xishuangbanna decreased from 12,326.49 km$^2$ in 2000 to 11,478.11 km$^2$ in 2010 [43]. In our results, the forest reduction rate of Xishuangbanna from 2001 to 2010 was 4.3%, and the main area where forest is reduced is east of Xishuangbanna (Table 3h).

Many studies have shown that Shanghai has experienced obvious urbanization and urban land expansion [44–46]. In these studies, Cui et al. [44] showed that the percentage of change for building land in Shanghai during 1980–2008 was 221.5%, and Feng et al. [45] showed that the high-density built-up areas increased from 5% in 1995 to 12% in 2015. In

our results, construction in Shanghai expanded at a 22.2% growth rate from 2001 to 2008 and was concentrated in central Shanghai. Shi et al. [47] mapped the impervious surface change in Wuhan using Landsat time-series datasets from 1987 to 2016, and their results show that the impervious surface area increased from 5.27% of the total land area in 2000 to 9.25% in 2011, growing at an annual average rate of 6.25%. In our results, the average value of China-LCF$_{MCD-CCI}$ in the construction of Wuhan increased from 0.06 in 2001 to 0.09 in 2013, increasing with an annual average rate of 2.9%.

**Table 3.** Comparison with existing studies.

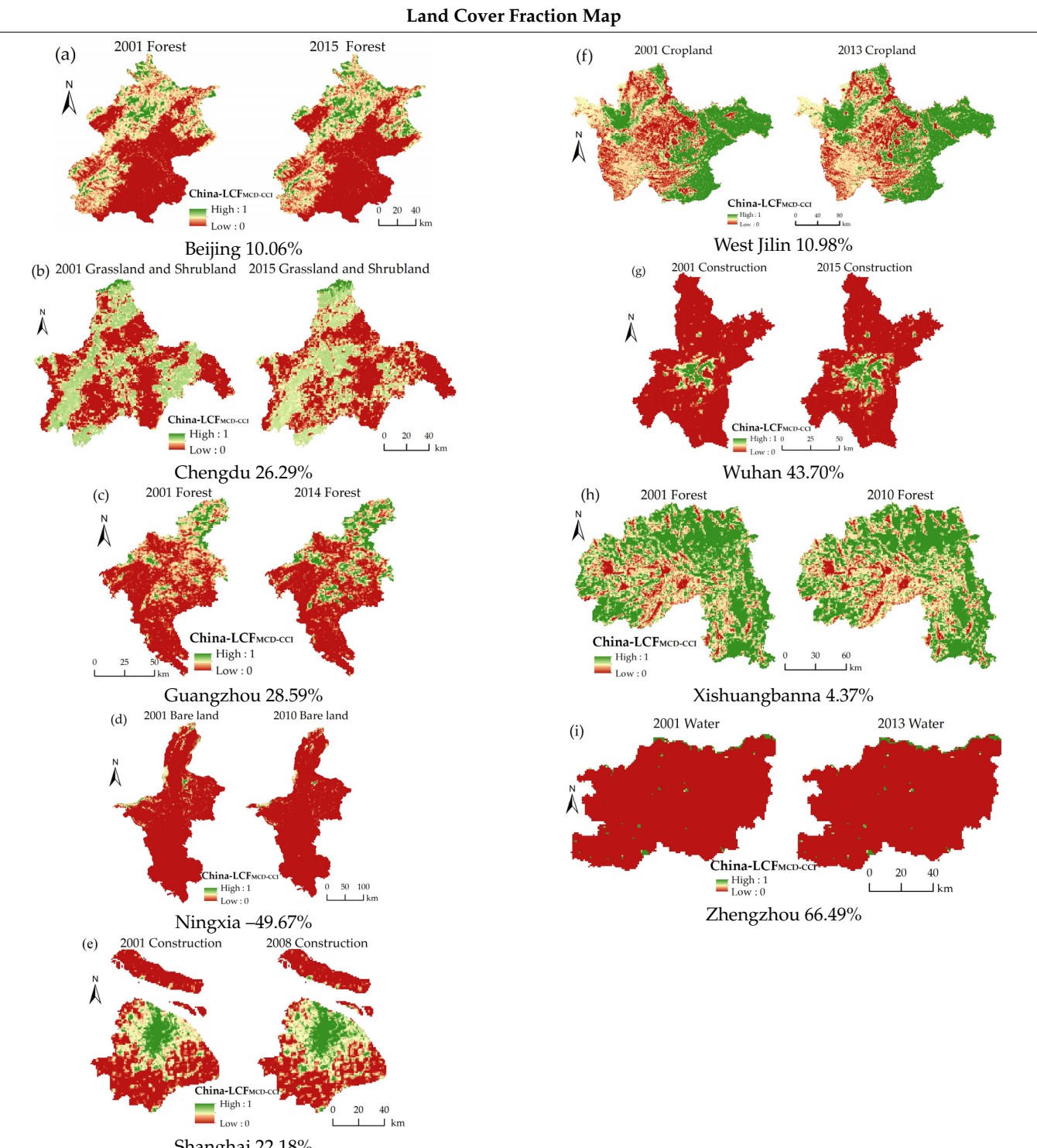

Li et al. [48] obtained data on the land cover change from 2000 to 2015 through remote sensing images of Chengdu and found that grassland was reduced by 7.5%. However, in our study, the growth of grassland and shrubland in Chengdu was 26.3%, with obvious increases in the middle of Chengdu. The reason for this difference may be that the study of [48] did not include shrubland, resulting in low estimations of the change area and change rate. Cadavid Restrepo et al. [49] assessed and quantified temporal and spatial changes in land cover in the Ningxia Hui Autonomous Region and found that the proportion of bare land fell from 40.3% in 2000 to 26.2% in 2015. In our study, the bare land in Ningxia was reduced from 2001 to 2015, and the reduction rate was 49.7%, which was significantly higher than the result reported by [49]. The reduction of bare land occurred mainly in northern and central Ningxia (Table 3d). Li et al. explored the impact of land use change on the ecosystem service value in West Jilin from 1976 to 2013, and found that the conversion of grassland and wetland to cropland resulted in an increase of 45,500 hectares in cropland in West Jilin from 2000 to 2013 [50]. Our conclusions were that the growth rate of cropland in West Jilin was 11.0% from 2001 to 2013, and there was a significant increase in the western part of West Jilin. Mu et al. used remote sensing data to identify the relative influences of planning policies on trends in landscape patterns in Zhengzhou, and found that the area of land covered by water in Zhengzhou increased from 141.3 km$^2$ in 2000 to 198.0 km$^2$ in 2013 [51]. Our study concluded that the area of land in Zhengzhou covered by water increased by 66.5% between 2001 and 2013 and was mainly concentrated in the northern part of Zhengzhou. In general, our study had a good consistency compared with the previous studies.

## 5. Conclusions

We determined that the change in grassland and shrubland was the most stable and the change in cultivated land fluctuated the most by the result of data fusion. The fraction of forest in China increased by 2.96% between 2001 and 2015, while the fractions of grassland and shrubland, cropland, and bare land decreased by 1.09%, 0.05%, 3.04%, respectively. Our study shows that continued time series land cover datasets were important for identifying land cover change, and our product can provide new evidence for studying China's land cover change. Based on the advantage of the datasets, the China-LCF$_{\text{MCD-CCI}}$ presented the time series fraction of different land cover types in China from 2001 to 2015. By calculating the slope, the areas where the fraction of the land cover changes were obvious for different land cover types can be identified. Due to the limitations of the datasets, we were only able to perform a consistency analysis of the 2010 dataset. In addition, the spatial resolution of the China-LCF$_{\text{MCD-CCI}}$ is 1 km, which is relatively coarse. Therefore, more datasets are required in the land cover fraction analysis to enhance the accuracy and resolution.

**Author Contributions:** Conceptualization, X.W.; methodology, H.W.; validation, X.W. and Y.W.; investigation, H.W.; data curation, H.W.; writing—original draft preparation, H.W.; writing—review and editing, Y.L., L.C., and D.P. All authors have read and agreed to the published version of the manuscript.

**Funding:** This research was financially supported by the Second Tibetan Plateau Scientific Expedition and Research Program (2019QZKK0405), the National Natural Science Foundation of China (41991235, 41877523), and the Shandong Provincial Natural Science Foundation (ZR2019BD045).

**Data Availability Statement:** The dataset is available online at https://doi.org/10.6084/m9.figshare.13147928.v4.

**Conflicts of Interest:** The authors declare no conflict of interest.

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
