# Peer review of "China’s Land Cover Fraction Change during 2001–2015 Based on Remote Sensed Data Fusion between MCD12 and CCI-LC"

_remotesensing, doi:10.3390/rs13030341_

Round 1

Reviewer 1 Report

Dear Authors,

The manuscript:

China's land cover fraction change during 2001-2015 based on remote sensed data fusion between MCD12 and CCI-LC

is well written and well composed. The manuscript presents interesting topic and research results.

I have some concerns to the quality of text, which consists of lack of proper ending of Introduction (hypothesis and the objective) some language issues as well as form of writing. All suggestions are given in the main file.

Good luck in your scientific work; with best wishes,

Rev

Reviewer 2 Report

The manuscript "China's land cover fraction change during 2001-2015 based on remote sensed data fusion between MCD12 and CCI-LC" is within the scope of Remote Sensing. However, it needs to go through Major Revision before being considered for publication. I ask the authors to carefully check the comments below. - it was not clear to me what the research objectives are, what its hypotheses. Therefore, the authors must insert in the Abstract and at the end of the Introduction what are their hypotheses and what are their objectives clearly; - The introduction did not clearly demonstrate the importance of this study for the region. There are several similar works. What are the weaknesses of these works that this manuscript fills? - The item Material and Methods is very poorly organized. Authors must insert a map of the location of the study area; - I suggest to the authors to carry out a flowchart for all procedures performed in M&M. Great work is needed on this item to make it clearer to readers; - The authors carried out some statistical analyzes, but they are insufficient to reach the proposed objectives. I suggest that they include techniques such as main component analysis and non-parametric tests such as those by Man-kendall and Pettitt. The significance of the non-parametric tests can be inserted in Figure 5; - Authors must inform all software used to process images and statistical tests; - The conclusions are too long. I suggest that the authors readjust the conclusions to the proposed objectives, in a clear and objective way.

Reviewer 3 Report

This manuscript reports the changing of the China's land cover fraction during 2001-2015 based on remote sensed data fusion between MCD12 4 and CCI-LC. The terrestrial ecosystems are very important to mitigate the global climate change. So, accurately evaluating the trend in land cover fraction change remains relevant in the global warming and carbon cycles context. This subject appears very appropriate for the Remote sensing journal. I like this manuscript and find it novel, interesting and well crafted. The dataset is long, the calculated values and interpretation are very competent, again making this paper better than others.

2) Materials and methods

  1. I suggest to authors to introduce the following sections or subsections :

2.1. China and its land cover description

2.2. Uncertainty estimation approaches

  1. Line 197 : Why did the authors choice four different land cover to validate China-LCFMCD-CCI? Justify.

3) Results:

  1. Scientifically, I would know more about the uncertainties. They should be estimated for all calculated values. This already help to better appreciate the validity of results. What they think?

Overall, the manuscript needs a minor revision. My suggestion is that it can be published in the Journal of Remote sensing after minor revisions.
